# Stability of *Bacillus* and *Enterococcus faecium* 669 Probiotic Strains When Added to Different Feed Matrices Used in Dairy Production

**DOI:** 10.3390/ani13142350

**Published:** 2023-07-19

**Authors:** Bruno I. Cappellozza, Audrey Segura, Nina Milora, Christel Galschioet, Mette Schjelde, Giuseppe Copani

**Affiliations:** Chr. Hansen Animal and Plant Health & Nutrition, Bøge Allé 10-12, 2970 Hørsholm, Denmark

**Keywords:** *Bacillus*, *Enterococcus faecium*, milk replacer, mineral–vitamin premix, pellet, stability

## Abstract

**Simple Summary:**

Direct-fed microbials (DFM), or probiotics, are live bacteria that confer health benefits to dairy animals and to bring such benefits, the bacteria must survive the feed preparation process. Hence, our objective was to evaluate the stability of *Bacillus* spp. and *Enterococcus faecium* 669 when included in different feed matrices. Three Exp. evaluated the recovery of a *Bacillus*-based (BOVACILLUS^TM^) on pelleted feed, mineral premix, and milk replacer. Another three Exp. evaluated the recovery of a DFM containing *E. faecium* 669 (LACTIFERM^®^) following its inclusion in a mineral premix and milk replacer prepared under different conditions. Overall, the recoveries of both *Bacillus* spores and *E. faecium* were stable and not impacted by the temperature used during the pelleting process, inclusion into a mineral–vitamin premix over 12 months, as well as when included into different milk replacer preparations. Altogether, the results from the present Exp. demonstrate, for the first time, the stability of a *Bacillus*-based and *E. faecium*-based DFM when included in different feed matrices often fed to the dairy cattle herd. These data provide novel information to the dairy industry in how to evaluate one of the features of a DFM (stability), as different results may be seen depending on the strains used.

**Abstract:**

Few data are available evaluating the stability of direct-fed microbials (DFM) following their inclusion in different feed matrices. Therefore, six Exp. evaluated the recovery of bacilli spores (BOVACILLUS^TM^; Exp. 1 to 3) and an *Enterococcus faecium* DFM (LACTIFERM^®^; Exp. 4 to 6) when included in different feed preparations. The *Bacillus*-based DFM was included into pelleted feed prepared in different temperatures (75 to 95 °C), whereas both DFM were assessed in premix and milk replacer preparations. *Bacillus* spores and *E. faecium* recovery was evaluated through standard methodologies and data were reported as log10 colony forming units/gram of feed. The recovery of *Bacillus* spores was within the expected range and was not impacted by the temperature of pellet preparation (Exp. 1). Bacilli recovery was also stable up to 12 months in the premix and was not impacted by the temperature of milk replacer preparation. Regarding the Exp. with *E. faecium* (Exp. 4 to 6), its recoveries in the mineral premix and milk powder did not differ from T0 and were not impacted by the conditions of milk replacer preparation. These data are novel and demonstrate the stability of a *Bacillus*-based and an *E. faecium*-based DFM when included in different feed matrices often used in dairy production.

## 1. Introduction

In recent years, direct-fed microbials (DFM) have been gaining more space into ruminant production settings, as these confer health benefits to the dairy animal [1,2,3,4], regardless of its age or category, while also improving nutrient utilization [5,6], overall performance of the herd [7,8], and the profitability of the dairy operations. Moreover, DFMs have been recognized as a potential alternative to antibiotic feeding in livestock production. Direct-fed microbials are often split into bacteria- or live yeast-based microorganisms and among the bacterial strains commonly used as DFMs for cattle, special consideration has been given to *Lactobacillus* spp., *Propionibacterium freudenreichii*, *Enterococcus faecium*, *Bifidobacterium* spp., and *Bacillus* spp. However, it is important to highlight that not all the aforementioned bacterial strains are able to survive and thrive in the same environments, which include different feed types, moisture, and high temperatures during feed preparation. Therefore, if the strains used in a commercial DFM are not able to survive the process of feed preparation, the health benefits expected may not be achieved and the success of the DFM might be questioned. Nonetheless, to date and to the best of our knowledge, no studies have evaluated the stability of lactic acid-producing bacteria, such as *E. faecium*, when added to different feed matrices often included in dairy production systems (i.e., milk replacer and mineral–vitamin premix).

On the other hand, *Bacillus* spp. are spore-forming bacteria that can survive the most challenging conditions, such as extreme temperatures, presence/absence of oxygen, wide pH ranges, the presence of radiation and toxic compounds, presence of other feed molecules and/or additives, as well as nutrient depletion in the environment [9,10,11], as bacterial spores maintain their viability, but do not have metabolic activity. This last statement is important and key, as it shows that the *Bacillus* spp. are not being lost during the feed manufacturing process [10,11], becoming active only when consumed by the animal (calf, developing heifer, or cow). However, limited information is available regarding the stability and recovery of *Bacillus* spores following its mixture in different feed matrices often included in dairy production settings that employ different conditions during its preparation, such as temperature, salinity, and moisture.

Based on this rationale, we hypothesized that the recovery of *E. faecium* 669 and *Bacillus* spores following different feed preparations (pelletization, mineral–vitamin premix, and milk replacer) would be comparable and stable after inclusion in the different feed matrices. Hence, our objective was to evaluate the recovery of *Bacillus* spores following its mixture in pellets prepared under different temperatures (75 to 95 °C; Experiment 1), mineral–vitamin premix (Experiment 2), and milk replacer (Experiment 3), but also of *E. faecium* 669 following its inclusion into a mineral–vitamin premix (Experiment 4), as well as a milk replacer powder (Experiment 5) or milk replacer preparation (Experiment 6).

## 2. Materials and Methods

### 2.1. Experiments

#### 2.1.1. Exp. 1—Pellet Trial with *Bacillus* spp.

This experiment was conducted to evaluate the recovery of *Bacillus* spp. when included in pelleted feeds prepared under different temperatures (75, 85, and 95 °C). A supplement containing (as-fed) 29.0% wheat, 20.0% barley, 15.0% corn, 10.0% beet pulp, 10.0% rapeseed cake, 9.5% high-protein soybean, 3.8% soybean oil, 1.0% limestone, and 1.7% mineral–vitamin premix was prepared.

The *Bacillus*-based DFM contained a mixture of *B. licheniformis* and *B. subtilis* (BOVACILLUS^TM^; Chr. Hansen A/S, Hørsholm, Denmark) was added into the above supplement to simulate a dosage of 3 g/head per day (representative of 9.6 × 10^9^ colony forming units (CFU)/head per day) in a dairy cow consuming 20 kg of dry matter (DM) per day. Based on these calculations, the expected recovery of the *Bacillus* spores was 3.8 × 10^5^ CFU/gram of pellet, regardless of the temperature used during the pellet manufacturing.

In the pellet production facility, a mill and mixer were used for producing meal mixtures and are part of a semi-industrial feed plant with a nominal pelleting capacity of 5 ton/h. The mill is a Champion hammer mill equipped with a 37 kW motor and a 340° screen with an area of 0.43 m^2^. The tip speed of the hammer is 11 m/s, whereas the mill is equipped with 3.0 mm diameter holes. From the hammer mill, the ground raw material was pneumatically led to a 2500 L horizontal mixer. The diameter of the mixing rotor was 1000 mm and the speed of 27 rpm. The milled raw material was forcibly mixed for 10 min. Then, from the horizontal mixer, the finished meal mixture was carried via a bucket elevator to a container from which the meal material is used for pelleting trials in a small-scale milling plant, which comprised of a horizontal mixer, dosing screw, cascade mixer, a high-pressure boiler that provided the steam for 30 s, and a digital thermometer to measure the reached temperature of 75, 85, and 95 °C. The pellet press could hold 300 kg/h, whereas the batches produced for this Exp. were of 230 kg. Following the pellet press, samples were cooled at room temperature for 9–11 min prior to sampling.

For each temperature, samples were taken from the horizontal mixer and post-pelleted by manually sampling at least 10 different locations in the mixer or pellet storage bucket to ensure an adequate composited sample. Moreover, samples were analyzed in triplicates before and after the pellet preparation and reported as log10 CFU/gram of pellet. As aforementioned, 3.8 × 10^5^ CFU of *Bacillus* spores/gram of pellet was the expected recovery value based on the pre-pellet sample counts.

#### 2.1.2. Exp. 2—Mineral–Vitamin Premix Trial with *Bacillus* spp.

This experiment was conducted to evaluate the recovery of *Bacillus* spp. over a 12-month period when included in a commercial mineral–vitamin premix. The commercial mineral–vitamin premix (Vilomix, Mørke, Denmark) contained 88.0% DM, 16.5% calcium, 4.0% phosphorus, 9.0% sodium, 5.0% magnesium, 1.7% sulfur, 900,000 IU/kg of vitamin A, 90,000 IU/kg of vitamin D3, 4000 IU/kg of vitamin E, 1000 ppm of copper, 4000 ppm of manganese, 5000 ppm of zinc, 225 ppm of iodine, 50 ppm of selenium, and 25 ppm of cobalt.

The *Bacillus*-based DFM used herein was the same as reported in Exp. 1 (BOVACILLUS^TM^; Chr. Hansen A/S), which was included in the mineral–vitamin premix to simulate a dosage of 2 g/head per day (representative of 6.4 × 10^9^ CFU/head per day). Based on these calculations, the expected recovery of the *Bacillus* spores was 3.95 × 10^7^ CFU/gram of premix. The Exp. was conducted twice and samples were analyzed in duplicates immediately following the inclusion of the DFM (T0) and after 1 (T1), 3 (T3), 6 (T6), and 12 (T12) months at 25 °C post-DFM inclusion into the mineral–vitamin premix. The results were reported as log10 CFU/gram of premix.

#### 2.1.3. Exp. 3—Milk Replacer Trial with *Bacillus* spp.

This experiment was conducted to evaluate the recovery of *Bacillus* spp. over a 60 min period when mixed during the preparation of a milk replacer under different temperatures.

For this Exp., the milk replacer was dissolved in water at two different temperatures (37 or 50 °C) and then the *Bacillus*-based DFM (BOVACILLUS^TM^; Chr. Hansen A/S) was included at a rate 1.0 × 10^6^ CFU/mL of the milk replacer. Following this step, the milk replacer was incubated for 60 min at room temperature (25 °C). The Exp. Was conducted twice and samples were analyzed in duplicates immediately following the inclusion of the DFM (time 0) and then at 30 and 60 min post-DFM inclusion into the milk replacer and the results were reported as log10 CFU/gram of the milk replacer.

#### 2.1.4. Exp. 4—Mineral–Vitamin Premix Trial with *E. faecium* 669

The same commercial mineral–vitamin premix used in Exp. 2 was also evaluated here (Vilomix, Mørke, Denmark), but in a different timing and batch. The *E. faecium* 669 DFM used herein (LACTIFERM^®^; Chr. Hansen A/S) has been approved for the feeding of calves and veal calves up to 6 months of age in the EU [12]. The DFM was added to the premix at a dosage to be representative of 3.0 × 10^9^ CFU of *E. faecium* 669/head per day. Based on these calculations, the expected recovery of the *E. faecium* 669 was 4.2 × 10^7^ CFU/gram of premix.

The Exp. was conducted twice, samples were stored in plastic bags and analyzed in duplicates immediately following the inclusion of the DFM (T0) and after 1 (T1), 3 (T3), 6 (T6), and 12 (T12) months at 25 °C post-DFM inclusion into the mineral–vitamin premix. The results were reported as log10 CFU/gram of the premix.

#### 2.1.5. Exp. 5—Milk Powder Trial with *E. faecium* 669

This experiment was conducted to evaluate the recovery of *E. faecium* 669 over a 60 min period when mixed in milk powder under different temperatures of preparation and storage. For this Exp., the milk powder contained (per 100 g) 509 kcal of energy, 27 g of fat, 56 g of carbohydrate, and 10 g of protein. The milk powder was dissolved in water that was heated to 37 °C (condition 1) or 50 °C (conditions 2 and 3) and then *E. faecium* 669 was added at 2.50 × 10^6^ CFU/g of the milk powder. Following this step, the preparation containing the milk powder and *E. faecium* 669 was stored for up to 60 min at 37 °C (condition 1), kept at 50°C (condition 2), or kept at 25 °C (room temperature; condition 3). The Exp. was conducted twice and samples were analyzed in duplicates immediately following the inclusion of the DFM (time 0) and then at 15, 30, 45, and 60 min post-DFM inclusion into the preparation, and the results were reported as log10 CFU/gram of the milk replacer.

#### 2.1.6. Exp. 6—Milk Replacer Trial with *E. faecium* 669

This experiment was conducted to evaluate the recovery of *E. faecium* 669 in a commercial milk replacer (Vilomilk-50, Mørke, Denmark) containing 50.0% skimmed milk powder, 20.0% whey powder, 18.0% vegetable fat, 4.0% wheat gluten, 3.0% aromas, 2.5% wheat starch, 1.0% wheat flour, 1.0% minerals, and 0.5% vitamins. The composition of the milk replacer was 22.0% crude protein, 18.0% fat, 6.9% ash, 0.98% calcium, 0.70% phosphorus, and 0.37% sodium.

The DFM was included in the milk replacer at a dosage to be representative of 2.5 × 10^9^ CFU/head per day. Based on these calculations, the expected recovery of *E. faecium* 669 was 8.65 × 10^6^ CFU/gram of the premix. The Exp. was conducted twice and samples were analyzed in duplicates immediately following the inclusion of the DFM (T0) and after 1 (T1), 3 (T3), 6 (T6), and 12 (T12) months at 25 °C post-DFM inclusion into the milk replacer. The results were reported as log10 CFU/gram of the milk replacer.

### 2.2. Counts Methodologies

#### 2.2.1. *Bacillus* Spore

Following the preparation of the feed matrices, the recovery of *Bacillus licheniformis* and *Bacillus subtilis* was performed according to the methodology based on EN-15784:2021 [13]. Briefly, the present method is a quantitative one, where the result was reported as CFU/g and then converted into log_10_ CFU/g. A known amount of sample was homogenized by stomaching with 0.2% sodium hydroxide (for Exp. 1) and Pepsal (for Exp. 2 and 3). Samples were heated at 80 °C for 10 min, cooled, and then serially diluted in the diluent. Appropriate dilutions were then spread on the surface of tryptone soy agar (TSA) plates. Lastly, after aerobic incubation for 16–24 h at 37 °C, colonies were counted.

#### 2.2.2. *Enterococcus faecium* 669

Following the preparation of the feed matrices, the recovery of *E. faecium* was performed according to the methodology based on EN-15788:2021 [14]. Briefly, the present method is a quantitative one, where the result was reported as CFU/g and then converted into log10 CFU/g. A known amount of *E. faecium* 669 sample was homogenized by stomaching with a diluent. Samples were heated at 80 °C for 10 min, cooled, and then serially diluted in the diluent. Appropriate dilutions were then spread on the surface of bile aesculin-azide (BAA) plates. Lastly, after aerobic incubation for 24 h at 37 °C, colonies were counted.

### 2.3. Statistical Analysis

In the present manuscript, data from all Exp. were analyzed with the SAS statistical software (version 9.4; SAS Inc., Cary, NC, USA) and, specifically, the MIXED procedure of SAS. For Exp. 1, samples were analyzed in triplicates, and temperature was considered a fixed effect. For Exp. 2, 4, and 6, results from T1 through T12 were compared against T0 to evaluate potential deviations in the counts of *Bacillus* spores (Exp. 2) or *E. faecium* 669 (Exp. 4 and 6) from the initial counts (T0). Lastly, in Exp. 3 and 5, the temperature or conditions used for the preparation (37 or 50 °C), storage of the milk replacer (25, 37, or 50 °C), and the hour were used to compare any differences against the data obtained on T0, if any. Additionally, a condition × hour interaction was also analyzed in Exp. 6.

Regardless of the Exp., all data were reported as least square means and significance was set at *p* ≤ 0.05, and tendencies were denoted if 0.05 < *p* ≤ 0.10.

## 3. Results

### 3.1. Exp. 1

Following the pellet manufacturing, the recovery of *Bacillus* spores was not impacted by the temperature used during pellet preparation (*p* = 0.25; Table 1). An additional analysis was performed to evaluate the deviation from the expected *Bacillus* spore counts and no differences were observed herein (*p* = 0.21; 100.1, 100.3, and 101.1% for 75 °C, 85 °C, and 95 °C, respectively; SEM = 5.5).

### 3.2. Exp. 2

No differences were observed when the data post-DFM inclusion in the premix were compared to T0 (*p* ≥ 0.17; Table 2). Moreover, when comparisons were performed among the other time points (T1 through T12), no differences were observed (*p* ≥ 0.06).

### 3.3. Exp. 3

Following the preparation of the milk replacer, no differences in *Bacillus* spore recovery were observed due to the temperature used for the dilution of the milk replacer (*p* = 0.62) or the time post-milk replacer preparation and *Bacillus* spp. inclusion (*p* = 0.68; Table 3).

### 3.4. Exp. 4

Following the inclusion of *E. faecium* 669 in the commercial mineral–vitamin premix, no differences were observed in its recovery due to the storage time (*p* = 0.25; Table 4).

### 3.5. Exp. 5

When the data post-DFM inclusion in the different conditions of milk powder preparation and storage were analyzed, no differences were observed regarding the mean recovery (log_10_ CFU; *p* = 0.17; Table 5) or the condition × hour interaction (*p* = 0.36). Moreover, Figure 1A–C report the counts of *E. faecium* 669 in the different conditions evaluated herein.

### 3.6. Exp. 6

Following the preparation of the milk replacer, no differences in *E. faecium* 669 recovery were observed (*p* = 0.22; Table 6).

## 4. Discussion

Direct-fed microbials, also known as probiotics, are beneficial bacteria that support the health of the host, modulating the microbiome of the gastrointestinal tract (GIT), inhibiting the damaging effects of potentially harmful bacteria [3,4,15,16], and supporting the immune reactions of the host [17]. However, in order to provide such benefits on health and ultimately on the performance of livestock animals, the bacterial strains used in commercially available DFM products must survive and drive the processes of feed preparation, which may include but are not limited to, low and high temperatures, presence of macro and trace minerals and high moisture, as well as long-term storage. Besides surviving the feed preparations, bacterial strains must also maintain their viability during the transit through the GIT of the ruminant animals, meaning that it will encounter changes in the type, quantity, and availability of substrates/nutrients, different pH values, competition from other bacteria (commensal or not) for nutrients and space, presence of bile salts, oxygen, and so on [18]. It is also noteworthy that it was not the goal of the present experiment to compare the recovery of different strains when included in the same feed matrix, so isolated and single analyses were performed across the different Exp. conducted here.

Most of the bacterial strains, such as lactic acid-producing bacteria (LAPB), require an additional coating/protective process to maintain their viability before their introduction into different supplements [19,20,21]. However, it is important to note that most of the studies performed with these bacteria have been performed under the conditions for human foods, which may not be representative of ruminant, and more specifically, dairy production settings. Hence, we evaluated the stability and recovery of *E. faecium* (LACTIFERM^®^), a well-known and heavily used bacterial strain for livestock [22] when mixed using different feed matrices and different conditions. Based on the results reported herein, *E. faecium* 669 was stable over time (Exp. 4 and 6) even when exposed to different temperature conditions during milk replacer preparation and storage (Exp. 5), ensuring the benefits often observed in previous in vivo Exp. from our research group [8,23]. Experiments such as those reported here provide the foundation of understanding the features of different bacterial strains included in commercial products, demonstrating its stability and guidance of utilization in commercial settings. For example, the fact that *E. faecium* 669 was stable over a long period of time following milk replacer preparation under different conditions is important and noteworthy, as it has been approved for use in pre-weaning and veal calves in the European Union [13].

Conversely to LAPB, other bacterial strains, such as *Bacillus* spp., present a natural resistance to several of the challenging factors, given its spore-forming feature [10,24]. More specifically, the spore-forming ability of *Bacillus* spp. ensures its full viability following long-term storage [24]. Moreover, bacilli spores can survive the challenging, low pH of the abomasum, reaching the small intestine at full viability to exert its beneficial effects on the host [25]. Given all these features and its enzyme-producing ability [26], *Bacillus* spp. have been used as DFM for livestock species, supporting the health and improving nutrient utilization [6,27] and performance of ruminants, including beef animals [7,28], dairy calves [29,30,31], and dairy cows [32,33,34]. However, to the best of our knowledge, minimal data from previous research have been published evaluating the stability and recovery of *Bacillus* spores in different feed matrices often used in dairy production systems, such as pelleted feed, mineral–vitamin premix, and a milk replacer. Therefore, these Exp. evaluated the recovery of spores included into a *Bacillus*-based DFM (BOVACILLUS^TM^) containing a mixture of *B. licheniformis* and *B. subtilis* when included in a pelleted feed (high-temperature; Exp. 1), commercial mineral–vitamin premix (presence of macro and trace minerals; Exp. 2), and a commercial milk replacer (high-temperature and moisture; Exp. 3).

Bacilli endospores are known to be highly resistant to physicochemical stress during feed production and storage, such as high pellet temperature, pressure, and shear forces [24]. Previous research has demonstrated that *Bacillus* spores are resistant to a pelleting temperature of up to 90 °C with over 90% of spores remaining viable in feed samples from poultry [35]. In Exp. 1, using temperatures that ranged from 75 to 95 °C yielded the projected recovery of *Bacillus* spores in pelleted feed for dairy cows. These data support the thermostability of *Bacillus* spores, adding an important feature to these strains to be used as a reliable source of DFM for ruminants.

In Exp. 2., timepoints were taken over a 12-month period to assess the recovery of *Bacillus* spores in a commercial mineral–vitamin premix when maintained in a room temperature environment (25 °C). Over time, no differences were observed among T1–T12 vs. T0, demonstrating the viability and resilience of this commercial DFM (BOVACILLUS^TM^) when added to a commercial mineral–vitamin premix containing different macro and trace minerals. Lastly, in Exp. 3, the recovery of *Bacillus* spores was assessed up to 60 min post-DFM (BOVACILLUS^TM^) mixture and milk replacer preparation under two different temperatures (37 and 50 °C) often used in the industry and previously reported by others [36]. The recovery of the *Bacillus* spores was not impacted by the temperature of milk replacer preparation nor by the time points assessed herein, demonstrating, once again, its stability in an important feed matrix to the dairy industry.

Nonetheless and lastly, it is important for the personnel involved in the dairy production chain (industry, farmers, nutritionists, and veterinarians) to recognize the differences among the genera, species, and strains of bacteria being used in different research efforts, as differences in these features impact the observed results, or lack of such. For other feed additives, such as yeast, ionophores, and non-ionophores, the differences between strains have been demonstrated, impacting the overall performance of ruminants [37,38,39].

## 5. Conclusions

In summary, our data demonstrate the stability of two different probiotic strains and products (*Bacillus*-based containing *Bacillus licheniformis* and *B. subtilis* (BOVACILLUS^TM^) and an *Enterococcus faecium* 669 (LACTIIFERM^®^)) when included in feed supplements prepared under different conditions. In fact, *Bacillus* spore recovery was stable in pelleted feed prepared using temperatures up to 95 °C, in a commercial mineral–vitamin premix containing different macro and trace minerals up to 12 months post-DFM inclusion, and in a milk replacer that was prepared under two different temperatures (37 and 50 °C). Lastly, *E. faecium* recovery was also stable when included in a commercial mineral–vitamin premix, commercial milk replacer, and when included into a mixture of milk powder under different conditions and storage. These data provide novel information that could help and improve the understanding of bacterial strains used as DFMs in dairy production.

## Figures and Tables

**Figure 1 animals-13-02350-f001:**
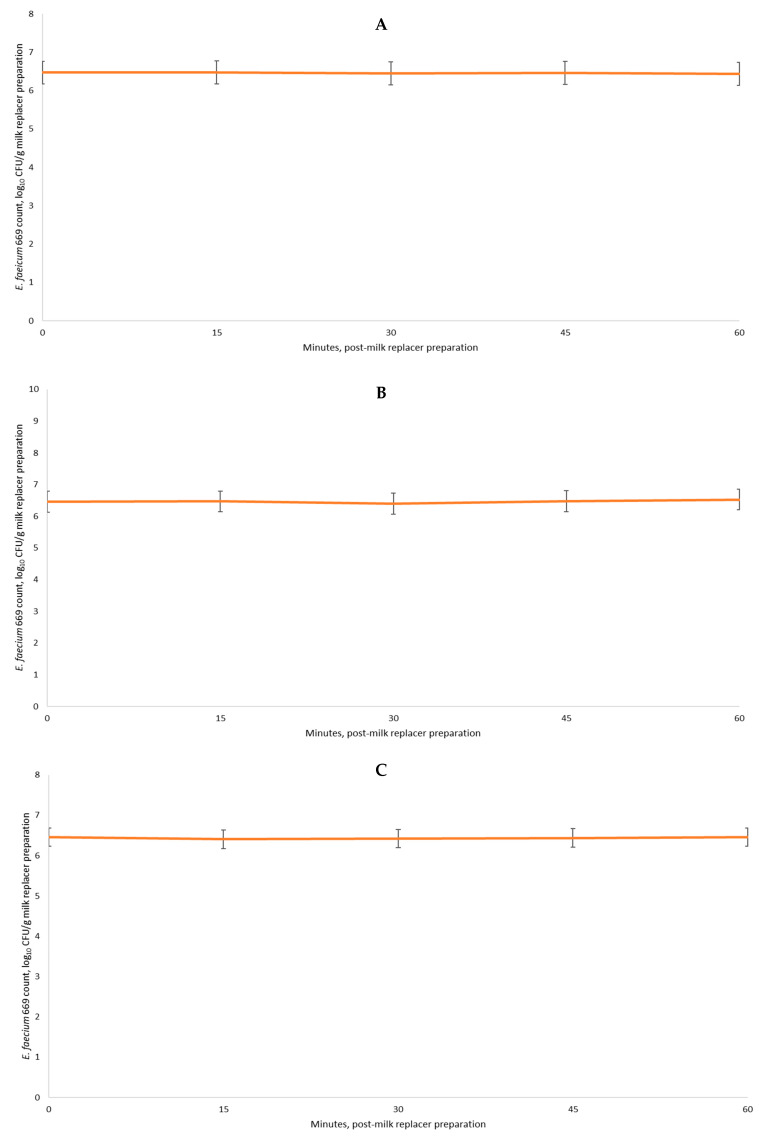
Recovery (mean ± SEM) of *E. faecium* 669 following its mixture with the milk powder under conditions 1 (**A**), 2 (**B**), and 3 (**C**). Within each condition, no overall mean differences were observed on counts (*p* > 0.10) or versus T0 (*p* > 0.12).

**Table 1 animals-13-02350-t001:** Recovery (mean ± SEM) of *Bacillus* spores following pellet manufacturing under three different temperatures (75, 85, and 95 °C). No temperature effect was observed in the observed recovery of *Bacillus* spp. (*p* = 0.25) ^1,2^.

Temperature, °C	Expected, log_10_ CFU/gram of Pellet	Observed, log_10_ CFU/gram of Pellet
75	5.58	5.58 ± 0.025
85	5.58	5.58 ± 0.025
95	5.58	5.64 ± 0.025

^1^ Samples were analyzed in triplicates; ^2^ *Bacillus*-based direct-fed microbials (DFM) consisted of a mixture of *B. licheniformis* and *B. subtilis* (BOVACILLUS^TM^; Chr. Hansen A/S, Hørsholm, Denmark).

**Table 2 animals-13-02350-t002:** Recovery (mean ± SEM) of *Bacillus* spores following its inclusion in the mineral–vitamin premix over a 12-month period. No overall differences were observed due to the month of *Bacillus* spp. assessment (*p* = 0.21) ^1,2^.

Time, Months	Observed, log_10_ CFU/gram of Premix	*p* = (Versus T0)
T0	7.60 ± 0.029	--
T1	7.56 ± 0.029	0.44
T3	7.66 ± 0.029	0.22
T6	7.60 ± 0.029	0.98
T12	7.51 ± 0.041	0.17

^1^ Samples were analyzed in duplicates; ^2^ *Bacillus*-based direct-fed microbials (DFM) consisted of a mixture of *B. licheniformis* and *B. subtilis* (BOVACILLUS^TM^; Chr. Hansen A/S, Hørsholm, Denmark).

**Table 3 animals-13-02350-t003:** Recovery (mean ± SEM) of *Bacillus* spores following its inclusion in the milk replacer prepared at either 37 or 50 °C and evaluated for up to 60 min. No overall differences were observed due to the temperature of preparation (*p* = 0.62) or timing of *Bacillus* spp. recovery (in minutes; *p* = 0.68) ^1,2^.

Item	Observed, log_10_ CFU/gram of Milk Replacer
Temperature, °C	
37	6.01 ± 0.021
50	6.00 ± 0.021
Time, minutes	
T0	6.02 ± 0.025
T30	6.02 ± 0.025
T60	5.99 ± 0.025

^1^ Samples were analyzed in duplicates; ^2^ *Bacillus*-based direct-fed microbials (DFM) consisted of a mixture of *B. licheniformis* and *B. subtilis* (BOVACILLUS^TM^; Chr. Hansen A/S, Hørsholm, Denmark).

**Table 4 animals-13-02350-t004:** Recovery (mean ± SEM) of *E. faecium* 669 following its inclusion in the mineral–vitamin premix over a 12-month period. No overall differences were observed due to the month of *E. faecium* 669 assessment (*p* = 0.25) ^1^.

Time, Months	Observed, log_10_ CFU/gram of Premix	*p* = (Versus T0)
T0	7.61 ± 0.054	--
T1	7.64 ± 0.054	0.64
T3	7.75 ± 0.054	0.13
T6	7.63 ± 0.054	0.74
T12	7.54 ± 0.054	0.42

^1^ Samples were analyzed in duplicates.

**Table 5 animals-13-02350-t005:** Recovery (mean ± SEM) of *E. faecium* 669 following its mixture with the milk powder under different conditions. No overall differences were observed due to the condition of the preparation containing *E. faecium* 669 (*p* = 0.17) ^1^.

Item	Observed, log_10_ CFU/gram of Milk Replacer Preparation
Condition ^2^	
1	6.46 ± 0.018
2	6.47 ± 0.018
3	6.44 ± 0.018

^1^ Samples were analyzed in duplicates; ^2^ Condition 1 = milk powder dissolved at 37 °C with the milk powder and DFM mixture stored at 37 °C; Condition 2 = milk powder dissolved at 50 °C with the milk powder and DFM mixture stored at 50 °C; Condition 3 = milk powder dissolved at 50 °C with the milk powder and DFM mixture stored at 25 °C.

**Table 6 animals-13-02350-t006:** Recovery (mean ± SEM) of *E. faecium* 669 following its inclusion in the milk replacer stored over a 12-month period. No overall differences were observed due to the month of *E. faecium* 669 assessment (*p* = 0.22) ^1^.

Time, Months	Observed, log_10_ CFU/gram of Milk Replacer	*p* = (Versus T0)
T0	6.95 ± 0.027	--
T1	7.00 ± 0.027	0.27
T3	7.00 ± 0.027	0.27
T6	6.95 ± 0.027	1.00
T12	6.90 ± 0.027	0.27

^1^ Samples were analyzed in duplicates.

## Data Availability

Not applicable.

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
