# Peer review of "Stability of *Bacillus* and *Enterococcus faecium* 669 Probiotic Strains When Added to Different Feed Matrices Used in Dairy Production"

_animals, 2023, doi:10.3390/ani13142350_

Round 1

Reviewer 1 Report

The manuscript is presented in a clear and easy to follow manner. I have comments and suggestions as follows:

Simple summary:

·         Line 20: the trade name (BOVACILLUSTM) is already given line 12; and the same for LACTIFERMTM. This is also repeated more than once in the ABSTRACT, M&M etc.  

·         Line 23: …novel guidance and confidence?? It is possible that the results presented may imply beneficial knowledge to the dairy industry , but “..novel guidance and confidence” sounds like an overkill here.

·         Line 23: “…how to select and choose a DFM”. The presented work did not involve any alternative feed additives but tested the stability of the probiotic strains. So, there is nothing to select and choose from.

·         Line 23: “…for use in dairy operations” instead of  “to be used in your dairy operation” ?

Based on the above comments, the last sentence of the simple summary should be re-written.

Abstract:

·         Line 26:   “and “ instead of “or”

·         Line 35: “…(CFU) per gram product”? Please be clear, here and elsewhere in the manuscript.

·         Line 37: T0 has not been defined.

·         Line 39:  … (Exp. 4 to 6); delete p.

·         Line 42: trade name again

·         Line 43: This was not a feasibility and efficacy test as far as I understand; this is recovery and stability test, and hence the conclusion should focus on the latter.

Introduction

The introduction is way too brief and shallow on the state of art.

·         Line 53: … survive and thrive? Survival comes first. (Check this also in the first paragraph of the discussion section).

·         Line 61: …their stability

·         Lines 69 to 71: Exp.6  is missing.

M&M

The sectioning in M&M was a bit extensive and mixed too.  The 6 experiments could have easily come under subsection “2.1 Experiments” and taken as either separate paragraphs or in Subsubsection (e.g. 2.1.1 Experiment 1; 2.1.2 Experiment 2.; etc). With this, subsections 2.4 and 2.8 (Bacillus spore counts  and Enterococcus faecium 669 counts) would come under subsection 2.2. Counts, followed by 2.3 Statistical analysis. As it is written now, subsections 2.4 and 2.8 get equal ranking as the experiments when in fact they are subsubsections.  

Specifics in M&M:

·         Line 84: at what expected recovery rate or %? (70, 80, 90%?). Please be clear.

·         Line 139: provide specifics on the diluent.

·         Line 168:  Do you really need this here: (LACTIFERM® ; Chr. Hansen A/S). e.g., You don’t use it in Line 155 above.

·         Line 190: delete “In the present experiment”. This is obvious. “Data from all experiments….” And here, avoid Exp. unless you refer to specific experiment, e.g. Exp.x)

·         Line 197: T0 as defined before in Lines 155 & 177.

Results:

·         Line 206: …117.3%? Rough calculation from the table doesn’t say so. Please check.

·         Line 257 to 262: Fig 1: Could you take a common x-axis title and classification? And show tick marks for x-axis.

Discussion

·         Lines 273 and 297: Refs 16 and 23 are not published works and cannot be cited as done here.

·         Line 276: ….thrive and survive.. see my earlier comment

·         Line 279: …their viability?

·         Line 285: Exp. see my earlier comments.

·         Line 290: delete …”readily and entirely.”

·         Line 337: Replace “as genera, species, and strain” with “these”??

Author Contributions:

·         Line 361: delete  ”Please turn to the CRediT taxonomy for the term explanation. Authorship must be limited to those who  have contributed substantially to the work reported.”

Informed Consent Statement

·         Line 364:Informed consent was obtained from all subjects involved in this study. “ should have been stated as “Not applicable”.

Author Response

Dear Reviewer,

Thanks for your comments and suggestions. We applied all the changes and replied to your comments, as attached.

Kind regards,

Bruno Cappellozza

Reviewer 2 Report

The title is too simple, maybe improved clarity and effectiveness

The abstract should include the experimental design. This helps readers understand how the study was conducted and provides context for the findings.

The introduction is too short and did not explain the importance of this type of innovation in the animal nutrition industry. it is crucial to provide sufficient context and explain the significance of the research topic, especially about the relevant industry or field. 

The experimental desing for each experiment needs to be described, including experimental n, post hoc test used, etc.

Did the mathematical assumptions test before the choice of statistical approach?

The conclusion needs to be clear and focus on the impact of the findings in the work. 

Line 342-346 This is a discussion. 

Author Response

Dear Reviewer, thanks for your comments and suggestions! We have applied and replied to all your points as attached.

Kind regards,
Bruno Cappellozza

Reviewer 3 Report

Simple Summary

It exceeds the word limit allowed by the paper (the maximum is 200).

The objective of the paper should be clear and, in my opinion, it is not.

Summary

The abstract also exceeds the maximum word limit set by the journal Animals, according to the instructions for the author (200 max). The abstract should also follow the following order: Background; Methods; Results and Discussion. The presented abstract does not present this structure, nor makes reference to the main objective. The conclusions are well defined and clear.

Introduction

The introduction is very concise, but manages to frame the work using recent references.

In the summary they indicate that they carry out 6 tests (L25), however, in the introduction they only refer to 5 tests.

Material and methods

One suggestion would be to schematise the experimental design of each trial and to tabulate the composition of the premixes.

In the statistician to use the "MIXED procedure of SAS" this should assume the assumption of normality, they do not make reference to this in the manuscript. Besides, they should indicate the Models used in each one of the tests.

The samples carried out only in duplicate remove the robustness of the statistical tests.

Results

None of the parameters analysed in the six trials obtained significant differences, and it is clear that the probiotics used do not change with the conditions presented.

Discussion

L271 - You have previously defined DFM, no need to write again.

After using abbreviations these should always be kept, you don't write once in full, again with abbreviation. Sometimes experiment appears sometimes Exp.

The discussion could focus more on the results obtained, referring to the results themselves in the discussion. For example they do not refer to the results of figure 1 in the discussion.

In table 4 and 6, although there are no significant differences, we can observe that at T3, the values reach a maximum and then decrease. Is there any explanation for this variation?

Conclusion

Clear and responds to the proposed objective. As a suggestion: they could add future perspectives.

Author Response

Dear Reviewer, thanks for reviewing the manuscript. Attached the comments and answers to your points.

Kind regards,

Bruno Cappellozza

Round 2

Reviewer 2 Report

The authors have improved the manuscript according to reviewer recommendations. 

Reviewer 3 Report

the authors accepted most of the suggestions and made the requested changes